# Integrative Genomic Tests in Clinical Oncology

**DOI:** 10.3390/ijms232113129

**Published:** 2022-10-28

**Authors:** Evgeny Imyanitov, Anna Sokolenko

**Affiliations:** 1Department of Tumor Growth Biology, N.N. Petrov Institute of Oncology, 197758 St. Petersburg, Russia; 2Department of Medical Genetics, St.-Petersburg Pediatric Medical University, 194100 St. Petersburg, Russia

**Keywords:** homologous repair deficiency, integrative tests, microsatellite instability, tumor mutation burden

## Abstract

Many clinical decisions in oncology practice rely on the presence or absence of an alteration in a single genetic locus, be it a pathogenic variant in a hereditary cancer gene or activating mutation in a drug target. In addition, there are integrative tests that produce continuous variables and evaluate complex characteristics of the entire tumor genome. Microsatellite instability (MSI) analysis identifies tumors with the accumulation of mutations in short repetitive nucleotide sequences. This procedure is utilized in Lynch syndrome diagnostic pipelines and for the selection of patients for immunotherapy. MSI analysis is well-established for colorectal malignancies, but its applications in other cancer types lack standardization and require additional research. Homologous repair deficiency (HRD) indicates tumor sensitivity to PARP inhibitors and some cytotoxic drugs. HRD-related “genomic scars” are manifested by a characteristic pattern of allelic imbalances, accumulation of deletions with flanking homology, and specific mutation signatures. The detection of the genetic consequences of HRD is particularly sophisticated and expensive, as it involves either whole genome sequencing (WGS) or the utilization of large next-generation sequencing (NGS) panels. Tumor mutation burden (TMB) can be determined by whole exome sequencing (WES) or middle-throughput NGS multigene testing. Although TMB is regarded as an agnostic indicator of tumor sensitivity to immunotherapy, the clinical utility of this test is proven only for a few cancer types.

## 1. Introduction

Many clinical decisions in cancer medicine are based on the results of molecular genetic tests. Most of these tests utilize a relatively straightforward approach, being focused on the alteration of a single gene. For example, many subjects undergo DNA analysis for hereditary cancer predisposition, and the detection of a germline pathogenic variant in a given gene is usually sufficient to assign the right diagnosis. While normal DNA is examined in patients with suspected familial cancer syndromes, the identification of potential drug targets generally relies on the detection of somatic mutations in tumor DNA. There are several dozen drugs that have been developed to counterbalance activated kinases. The administration of these compounds is often based on the identification of a druggable genetic lesion in cancer cells, be it kinase-activating point mutation, fusion, or gene amplification (Table 1). Nowadays, for the sake of convenience, the above analyses are often carried out by next-generation sequencing (NGS), which incorporates into a single run all potentially relevant cancer-related genes. However, the interpretation of NGS results is usually performed on a single-gene basis, with no consideration of other characteristics of the tumor genome [1].

There are several medically important DNA tests that evaluate the entire landscape of somatic mutations and provide grounds for the management of cancer disease (Table 1). These tests are generally more challenging, as they deal with continuous variables, and their performance and interpretation are significantly influenced by various technical issues and the choice of the threshold. Examples of these tests include microsatellite instability (MSI), tumor mutation burden (TMB), and homologous repair deficiency (HRD) (Table 1).

## 2. Microsatellite Instability

Microsatellites (short tandem repeats) are DNA segments that contain repetitive nucleotide sequences; the size of these repeats is usually in the range of 1–6 base pairs (Figure 1). Microsatellites are prone to errors arising during DNA replication, which are normally corrected by DNA mismatch repair (MMR). MSI was discovered at the beginning of the 1990s during the studies on Lynch syndrome, as well as investigations on the molecular pathogenesis of colorectal cancer (CRCs) [4,5,6,7]. MSI manifests as a change in the length of microsatellite sequences due to somatic mutations affecting tumor DNA.

MSI testing was initially utilized as a pre-screening test for the analysis of germline pathogenic variants in Lynch syndrome genes. Several factors complicate the performance of microsatellite analysis. Repetitive DNA sequences are often polymorphic in their size. Therefore, the early assays involved the comparison of tumor versus normal DNA. PCR-amplified microsatellites usually produce scattered bands on gel-electrophoresis; hence, the detection of small changes in their size is potentially error-prone (Figure 1). Furthermore, it appears that somewhat different biological mechanisms underlie instability in mononucleotide, dinucleotide, trinucleotide, and tetranucleotide repeats, so the optimal composition of microsatellite panels is a subject of debate [8].

MSI testing has significantly evolved in the first decade of this century. It was suggested to limit the analysis by mononucleotide repeats, as their alterations more specifically reflect MMR deficiency (MMR-D). Furthermore, several monomorphic mononucleotide repeats have been revealed, thus alleviating the need to analyze normal DNA. The most common panel includes five markers (BAT25, BAT26, NR21, NR24, NR27 (MONO27)), and MSI status is assigned to tumors demonstrating shifts for at least two out of five PCR fragments. The development of efficient immunohistochemical (IHC) assays for Lynch-related MMR proteins (MLH1, MSH2, MSH6, and PMS2) has enabled pathomorphological laboratories to carry out MMR-D testing. Most, although not all, inactivating mutations in MMR genes are protein-truncating and, therefore, result in the lack of production of the corresponding protein; furthermore, *MLH1* and *MSH2* alterations are accompanied by down-regulation of their heterodimeric partners. *MLH1* gene inactivation is usually manifested by the concomitant loss of MLH1 and PMS2 IHC staining; *MSH2* down-regulation results in the absence of expression of MSH2 and MSH6 proteins; *MSH6* and *PMS2* pathogenic mutations lead to the failure of IHC visualization of MSH6 and PMS2 proteins, respectively. In contrast to DNA-based assays, IHC can be applied to tumors with a low proportion between malignant and normal cells. However, there are some missense protein-inactivating mutations in the MMR genes, which do not result in the loss of IHC staining but lead to MSI [9].

Recent developments added significantly more complexity to MSI detection. In addition to the need to use MSI as a pre-screening test for Lynch syndrome, MSI turned out to be an agnostic marker for the efficacy of immune checkpoint inhibitors (ICIs). In fact, MSI is one of the mechanisms of highly increased TMB and results in an elevated number of tumor-specific antigens. Many sporadic tumors exhibit MSI due to the somatic inactivation of MMR machinery. Importantly, earlier studies aimed at developing reliable MSI assays focused mainly on MSI detection in colorectal tumors [10]. These tumors are characterized by a high rate of cell proliferation, and the concordance of the MSI testing by a conventional five-marker panel versus MLH1/MSH2/MSH6/PMS2 IHC is close to 100% [11]. Furthermore, almost all colorectal tumors arising in patients with Lynch syndrome are microsatellite unstable and have high TMB, and the same is true for sporadic CRCs arising due to *MLH1* promoter hypermethylation [8,10,12].

Investigations of non-colorectal tumors shaped several important nuances. It was revealed that the standard PCR panels often fail to reliably detect MSI in non-colorectal tumors, for example, in endometrial carcinomas, which are characterized by a particularly high rate of microsatellite instability [13]. Furthermore, there are instances of tumors with well-proven MMR-D status, often linked to Lynch syndrome, which do not accumulate a high number of mutations in microsatellites, possibly due to low proliferation rates [8,10,14,15,16]. Therefore, distinct criteria may be applied to the pre-screening test for hereditary cancer and to the assay aimed at the selection of patients for immunotherapy. It is safe to state that all available assays have not yet been sufficiently validated for the detection of MSI/MMR-D status in non-colorectal tumors.

There are two major avenues for the development of reliable MSI tests: the improvement of PCR-based platforms and the incorporation of MSI analysis into NGS diagnostic procedures (Figure 1). PCR is generally cheaper and has a lower turn-around time as compared to NGS; therefore, the development of an “agnostic” PCR test is an attractive goal. There are several commercial PCR platforms (e.g., OncoMate^TM^ MSI Dx Analysis System (Promega) or Idylla^TM^ MSI Assay (Biocartis)) as well as a multitude of laboratory-developed tests [10]. It has been reported that the standard five-marker panel (BAT25, BAT26, NR21, NR24, NR27 (MONO27)) may produce false-negative results in non-colorectal tumors, hence additional microsatellites are probably required for reliable MSI analysis in other cancer types. The change of microsatellite length in carcinomas with a low proliferation rate is less pronounced as compared to CRC. Therefore, a more precise determination of the PCR fragment size may be needed [13,17,18,19]. Some markers appear to be more informative than others [20]. For example, there are studies emphasizing the advantage of long microsatellite fragments [21].

Most NGS-based approaches for MSI detection utilize already existing NGS gene panels and involve bioinformatic algorithms aimed at the analysis of microsatellite DNA sequences [12,22,23,24,25,26,27,28,29,30,31,32]. Diagnostic and research NGS platforms are mainly focused on the analysis of exons of cancer-related genes; therefore, NGS-MSI tests are often enriched by coding gene sequences. This is a potential source of bias, given that many genes, and, consequently, their microsatellite tracks, are mutated in some tumor types but not altered in others [8,33].

The major issue for MSI analysis is the definition of the threshold. Some pan-cancer studies claim that this threshold varies across tumor types; however, even cancers with a low proportion of altered repetitive sequences can still be divided into microsatellite-stable and -unstable [34]. While this knowledge is definitely essential for the understanding of mechanisms of tumor development, the presumable clinical relevance of MSI/MMR-D appears to be more straightforward. As stated above, MSI/MMR-D status is potentially useful as a Lynch syndrome pre-screening test and as an indicator of high TMB, so the actual performance of newly developed assays has to be assessed towards these two endpoints.

## 3. Homologous Repair Deficiency (HRD)

Research on hereditary breast-ovarian cancer syndromes was initially viewed as a part of cancer prevention, as it focused mainly on the identification of women at-risk and the early detection of tumor lumps. Studies carried out in the first decade of this century revealed an unexpected vulnerability of BRCA1/2-driven adenocarcinomas. Almost all breast or ovarian cancers arising in *BRCA1/2* germline mutation carriers develop via the inactivation of the remaining allele of the involved gene. Both BRCA1 and BRCA2 play a crucial role in the repair of DNA double-strand breaks by homologous recombination (HR). Consequently, these hereditary tumors are selectively sensitive to the drugs, which induce DNA double-strand breaks (poly-(ADP-Ribose)-polymerase inhibitors (PARPi), cisplatin, mitomycin C, cyclophosphamide, etc.), while normal cells in patients with *BRCA1/2* mutation carriers tolerate this treatment due to retention of functional *BRCA1/2* allele [35].

Subsequent research revealed that many non-hereditary (sporadic) tumors share biological characteristics with BRCA1/2-driven cancers. This phenomenon has been initially defined as “BRCAness.” However, current literature more often utilizes the term “homologous repair deficiency (HRD).” The diagnosis of HRD status is important for the proper selection of therapeutic strategy. Therefore, there are intensive efforts aimed at the development of clinically useful HRD tests. These approaches include (1) comprehensive testing for germline pathogenic variants in *BRCA1/2* genes; (2) analysis of gene-inactivating germline variations in other members of the HR pathway; (3) identification of tumor-specific somatic mutations in *BRCA1/2* and similar genes; (4) complex assays aimed at the detection of chromosomal instability in tumor cells; (5) functional HRD assays; (6) analysis of the clinical course of cancer disease, i.e., platinum sensitivity of the tumor. All these tests have essential limitations.

*BRCA1/2* germline testing has been routinely established since the mid-1990s, although even now, its use is related to some difficulties. The identification of protein-truncating small mutations is a straightforward and reliable procedure normally achieved by next-generation sequencing (NGS). However, there are some missense mutations, which also lead to *BRCA1/2* inactivation, but their actual pathogenic significance is difficult to clarify. Furthermore, the existing multigene NGS assays have not been sufficiently validated for the detection of large gene rearrangements (LGRs), i.e., deletions or duplications of gross portions of the genes [36,37]. Some tumors in *BRCA1/2* germline mutation carriers still retain the functional copy of the involved gene; this is particularly characteristic for non-breast-ovarian cancer types [38,39]. Although current treatment standards for breast, ovarian, pancreatic and prostate cancers allow the administration of PARP inhibitors solely on the results of a germline test, ideal patient management would imply the subsequent analysis of tumor tissue aimed to prove the inactivation of the remaining *BRCA1/2* allele.

There are over a hundred genes implicated in DNA repair by homologous recombination, and several dozen of them are associated with increased cancer predisposition [40]. Some of these genes, e.g., *PALB2* or *RAD51* paralogs, appear to be related to the HRD phenotype, although important nuances need to be considered [41,42]. Somatic inactivation of the remaining gene allele is characteristic for the majority of, but not all, PALB2-driven tumors. Furthermore, in contrast to *BRCA1/2*, a normal copy of the *PALB2* is often affected not by deletion (“loss-of-heterozygosity,” LOH), but by small protein-truncating mutation [43]. Germline pathogenic variants in several other HR-related genes, for instance, *ATM* and *CHEK2*, certainly render increased cancer risk. However, tumors arising in mutation carriers often do not have signs of HRD and, therefore, are unlikely to be sensitive to PARPi or DNA-damaging cytotoxic drugs [42,44,45,46,47,48].

The analysis of HR mutations in tumor tissues has some potential advantages. It can reveal both germline pathogenic variants and somatically acquired mutations. In some instances, the analysis of tumors is logistically less complicated, as the biological samples can be obtained from a pathological archive, while the donation of blood DNA requires a patient visit. However, testing for somatic mutations has significant limitations, which have already been mentioned above. Most importantly, current standards do not require a thorough analysis of both alleles of the mutated gene. For instance, a phase III clinical trial of olaparib in prostate cancer involved tumors with identified mutations in *BRCA1*, *BRCA2*, *ATM*, *BRIP1*, *BARD1*, *CDK12*, *CHEK1*, *CHEK2*, *FANCL*, *PALB2*, *PPP2R2A*, *RAD51B*, *RAD51C*, *RAD51D*, and *RAD54L* genes, and the detection of a single mutational event was sufficient for the administration of this drug. These genes were collectively defined as HRR (homologous recombination repair) panels, and the corresponding NGS test was incorporated into the drug label [49]. In fact, somatic mutations in HR genes are not necessarily biallelic, and monoallelic alterations of the members of homologous repair pathways are not accompanied by HRD [38,42,50]. Another issue is the varying functional impact of HR gene alterations. Even *BRCA1*, which is obviously associated with the HRD phenotype in breast or ovarian cancer, does not necessarily undergo second-hit inactivation in pancreatic or prostate carcinomas [51,52]. The efficacy of olaparib in prostate cancer is primarily attributed to the presence of the *BRCA2* gene within the NGS-HRR panel, while many other genes included in the above list do not render sensitivity to this drug [47,49]. The detection of mutations in *ATM* and *CHEK2* genes is a particularly common finding; however, there are doubts regarding their HRD-associated role [42,45,46,47,48]. Importantly, biallelic inactivation of a given HR gene is not necessarily an “agnostic” indication, i.e., there are genes whose inactivation is associated with increased sensitivity to DNA damage in some but not all cancer types [53].

Homologous recombination is a high-fidelity mechanism of the repair of DNA double-strand breaks. HRD is accompanied by the activation of bypass mechanisms of genomic maintenance, which are error-prone and result in the emergence of characteristic patterns of genetic alterations. Widespread allelic imbalances are the best-established feature of HRD tumors. Some years ago, this chromosomal instability was detected by comparative genomic hybridization (CGH); currently, most laboratories utilize NGS assays targeted at single nucleotide polymorphisms (SNPs) distributed throughout the genome. Foundation Medicine offers LOH scoring to patients with high-grade serous ovarian cancer, where an index above 16% is interpreted as a sign of HRD. This assay demonstrated clinical significance in rucaparib clinical trials as well as in studies on platinum-based therapy [54,55,56]. The ovarian cancer study utilizing genome-wide LOH scoring revealed no correlation between the total size of LOH-affected regions and platinum sensitivity. Instead, BRCA1/2-mutated tumors and sporadic cancers responding to carboplatin/paclitaxel were characterized by a high number of evenly spread chromosomal breaks [50].

The most popular HRD NGS-based test has been developed by Myriad Genetics (Figure 1). It calculates a cumulative index for the HRD-associated chromosomal patterns, namely the appearance of widespread LOH, the presence of telomeric allelic imbalances (TAI), and the occurrence of large-scale transitions (LSTs). The analysis of BRCA1/2-driven cancers revealed that these tumors are characterized by accumulation of LOH, which are larger than 15 Mb in size but do not involve the entire chromosomes [57]. LSTs are defined as chromosomal breaks between adjacent regions of at least 10 Mb, i.e., they are represented by multiple gains and losses of genetic material encompassing relatively large genomic regions [58]. Gains and losses are particularly characteristic for telomeric regions of BRCA1/2-related cancers, and this feature is defined as telomeric allelic imbalance [59]. HRD analysis is based on the scoring of LOH, TAI, and LST. The HRD cut-off was arbitrarily estimated based on the capability of HRD assay to detect 95% of tumors associated with *BRCA1/2* mutation or *BRCA1* promoter methylation. This cut-off >/= 42 has been robustly validated in several breast and ovarian cancer trials involving PARPi or platinum therapy [60].

Tumors with germline *BRCA1/2* mutations usually demonstrate better sensitivity to platinum compounds or PARPi than sporadic carcinomas with high HRD or LOH scores, and these observations are generally consistent across studies [50,56,61,62,63]. On the other hand, there are some patients with scores below the accepted threshold who still obtain benefits from the therapy. Consequently, there are clinical studies searching for optimal cut-offs for discrimination between HR-deficient and proficient tumors [62,64,65]. Importantly, the threshold values have been calculated only for breast and ovarian cancer patients, and it is unclear whether these estimates are optimal for other cancer types.

There are computing algorithms that consider the entire complexity of genome-wide BRCA1/2-specific genetic alterations (Figure 1). While the above HRD scores consider only allelic imbalances, the HRDetect tool also evaluates the pattern of point mutations, an excess of deletions of genetic material with stretches of sequence homology at junctions (“microhomology”), and the profile of DNA rearrangements [66]. Microhomology-mediated end joining (MMEJ) is a compensatory DNA repair pathway that is activated in HR-deficient tumors and results in a significant accumulation of deletions with flanking homology [67]. This feature is particularly characteristic of the HRD phenotype; however, its analysis currently requires whole-genome sequencing (WGS) of tumor DNA [39,64]. There is a freely available computing tool, “Classifier of HOmologous Recombination Deficiency” (CHORD), which is aimed at the identification of HRD status through the interpretation of WGS data [41].

The existing HRD genomic tests require the use of large NGS panels and sophisticated bioinformatics algorithms. There are attempts to develop relatively simple and intuitively attractive surrogate markers for HRD. For example, the analysis of copy number variations in *MYC*, *CCNE1*, and several other loci, which can be performed by digital droplet PCR, provides a rapid and cost-efficient approximation of the genome-wide profile of allelic imbalances [50,65,68]. HR involves the interaction between RAD51 protein and DNA breaks; therefore, tumors with HRD are characterized by low IHC staining for RAD51 foci [69].

There are ex vivo assays utilizing the treatment of biopsied tumor tissue with DNA-damaging agents followed by the scoring of RAD51 foci. Homologous repair involves the recruitment of RAD51 protein to DNA double-strand breaks; consequently, cells with HRD do not demonstrate staining for RAD51-DNA complexes upon the induction of DNA lesions. RAD51 foci are intrinsically rare in tumors with low proliferation rate. Therefore, some RAD51 assays involve the visual selection of mitotically active cells by geminin or Ki-67 staining [60]. Functional tests have an obvious advantage, as they evaluate the genuine activity of HR and can be applied to both chemonaive and pretreated tumors. Indeed, HR proficiency is often rescued during therapy courses through the second mutation restoring the open reading frame of *BRCA1/2* or similar genes or by other bypass mechanisms [70]. The genomic scars described above remain predictive for HRD during the treatment course, even when the actual sensitivity to PARPi or platinum compounds is lost [50]. In contrast, functional RAD51 assays are capable of discriminating between HR deficiency and proficiency, irrespective of the genomic context. Interestingly, these assays may identify a subset of tumors that have evidence for HRD, despite the lack of an HRD-related mutation footprint [71]. It should be kept in mind that RAD51 tests are unable to identify HR-deficient tumors which have inactivating alterations in HR pathway located downstream to RAD51. The requirement for immediate laboratory analysis of viable tumor tissue is a strong disadvantage of functional tests [72,73].

Discrimination of HR-deficient and -proficient tumors is mainly needed for the upfront administration of DNA-damaging cytotoxic drugs or PARPi. For the time being, HRD assays are increasingly utilized for PARPi, while the use of platinum compounds remains largely empirical. PARPi indications for ovarian cancer are limited to platinum-sensitive disease, given almost complete cross-resistance between these two therapies. While some indications for the use of PARPi are based on the utilization of genetic tests for HRD, there are also FDA-approved scenarios when the clinical decision is entirely based on the prior response to platinum compounds [37,73,74,75].

HRD tests received an intensive promotion in the past due to the invention of novel drugs, such as olaparib, rucaparib, niraparib, and talazoparib. Platinum compounds and some other DNA double-strand break-inducing agents form a backbone for the therapy of a huge number of cancer patients. For example, carboplatin or cisplatin are given upfront to virtually all patients with high-grade serous ovarian cancer, with more than a third of them having no benefit from this approach. Current HRD tests allow the identification of tumors that have a low probability of responding to the standard therapy; these subjects are candidates for clinical trials involving alternative treatment strategies [50].

## 4. Tumor Mutation Burden (TMB)

Malignant transformation is a process mediated via the mutation-driven activation of oncogenes and the inactivation of suppressor genes. The number of “driver” genetic events necessary for the acquisition of a neoplastic phenotype is within a range of 2–10, being usually lower in leukemia and higher in common epithelial tumors. In addition to “driver” mutations, tumors always contain “passenger” somatic alterations of DNA sequences [76]. The total number of mutations, defined as “tumor mutation burden” (TMB), depends on the intensity of carcinogenic exposure, the activity of DNA repair mechanisms, and the number of errors incorporated during DNA replication. For example, high TMB is characteristic of smoking-induced lung cancer, ultraviolet-associated skin melanoma, tumors arising due to deficiency in DNA mismatch repair, POLD1/POLE-associated cancers, etc. In particular, microsatellite instability (see above) is a variant of high TMB. Non-synonymous somatic mutations may lead to the appearance of tumor-specific antigens (neoantigens); therefore, high TMB is potentially associated with increased tumor immunogenicity [77].

TMB was initially defined as the number of somatic missense mutations evaluated via the comparison of tumor vs. normal exome [78]. High TMB generally correlates with the efficacy of immunotherapy in tumors with high carcinogen-induced mutation loads, e.g., lung cancer and melanoma [78,79,80,81]. Exome analysis of paired tumor-normal tissues is not yet compatible with clinical routine. Therefore, most TMB tests are currently performed by NGS panels consisting of a few hundred genes [77]. It is essential to acknowledge that NGS multigene tests were originally developed to analyze actionable mutations, so they are composed of genes with a well-proven role in cancer pathogenesis. Therefore, while exome sequencing provides an unbiased overview of driver and passenger mutations, the results of NGS multigene tests are usually enriched by actionable genetic events. For example, virtually all NGS assays include *TP53*, *BRAF*, *KRAS*, *NRAS*, etc. genes, which are frequently mutated in particular tumor types but may not directly reflect the TMB status of a given tumor. Furthermore, while exome-based calculations of TMB usually consider several dozens or hundreds of somatic mutations, the size of NGS panels is 10–20 times smaller than exome, and the same applies to the number of identified genetic events. Consequently, multigene assays are reliable in detecting outliers, e.g., tumors with very high or very low TMB, but may lack precision in the analysis of tumors with intermediate TMB values [82].

There are some other simplifications related to the use of commercial NGS platforms [77]. In contrast to exome studies, most multigene tests score for both non-synonymous and synonymous mutations. Synonymous mutations constitute approximately a quarter of TMB; although they do not influence the immunogenicity of the tumor, they reflect the history of carcinogen exposure or the failure of DNA repair machinery [83]. Hence, the inclusion of synonymous mutations inflates the total mutation score and thus compensates for the low size of commercial NGS panels. Furthermore, synonymous mutations rarely play a role of driving events, so they may reflect TMB more properly than actionable alteration [84]. In addition to missense mutations, multigene tests consider small deletions and insertions. The majority of commercial NGS assays analyze only tumor tissue and do not require a comparison with normal DNA. Consequently, some unknown polymorphisms are likely to be interpreted as somatic mutations, and this risk is increased for ethnic groups with poorly characterized germline genetic diversity [77].

In addition to the technical caveats of TMB measurement, the clinical utility of TMB remains the subject of discussion. Exceptionally high TMB in carcinogen-unrelated tumors may be an indicator of hereditary cancer syndrome associated with the inactivation of *POLD1*, *POLE*, *MUTYH*, MMR (*MLH1*, *MSH2*, *MSH6*, *PMS2*), or some other genes, so in rare circumstances, TMB results may call for germline genetic testing [36]. TMB is regarded as an appropriate predictive marker for immunotherapy response in non-small cell lung cancer (NSCLC) and cutaneous melanoma [78,79,80,81]. However, its actual use in these tumor types is relatively uncommon. Smoking history is an efficient surrogate of high TMB in patients with NSCLC; in addition, PD-L1 expression testing is widely utilized to guide the use of inhibitors of immune checkpoints in lung cancer management [2,85]. Laboratory testing for conventional cutaneous melanoma relies mainly on *BRAF* testing, with the subsequent consideration of either inhibition of MAPK cascade or the use of immunotherapy, so the TMB testing is unlikely to influence the actual treatment strategy [86]. The situation is significantly more complex regarding other tumor types. NSCLC and melanoma studies revealed a provisional threshold of 10 mutations per megabase (Mb), which allows the prediction of high versus low probability of response to immunotherapy (Figure 1). These data were extrapolated to several other categories of malignancies, and the success of the KEYNOTE-158 trial led to the approval of the TMB as an agnostic indication. This approach received a significant amount of criticism. It was commented that only a few cancer types (endometrial, cervical, bladder) demonstrated a correlation between high TMB and efficacy of immune drugs, while other categories of tumors (anal carcinomas, gliomas) did not show this relationship [77,80,81,87,88]. McGrail et al. [80] revealed that the predictive value of high TMB is observed only for tumor types in which high neoantigen load is associated with increased CD8 lymphocyte infiltration. Some common cancer entities, e.g., breast and prostate cancers, showed no association between high TMB and lymphocyte infiltration and actually demonstrated an inverse relationship between TMB and the probability of tumor response to immune checkpoint blockade [80]. The clinically relevant threshold for TMB is unlikely to be the same across all tumor types, so the adjustment to the histological origin of cancer disease may be necessary [80,89]. As a consequence, ESMO recommendations limit the use of TMB to several cancer varieties, namely cervical cancer, salivary cancer, thyroid cancer, well-to-moderately differentiated neuroendocrine tumors, and vulvar cancer. This conclusion is based mainly on the results of the KEYNOTE-158 pembrolizumab clinical trial; however, the latter four tumor types were represented by only 22 patients in total, with only seven tumor responses observed [87]. In addition, the recommendations for TMB testing include small-cell lung cancer and endometrial cancer, but only if immunotherapy for these two entities is not available otherwise [90]. It has to be noticed that MSI/MMR-D is a common cause of high TMB in endometrial cancer; MSI diagnosis in uterine tumors is less standardized as compared to colorectal cancers (see above), and it remains to be established whether TMB determination renders significant added value to the MSI/MMR-D testing in this tumor group.

The quantitation of the total number of mutations may be an oversimplification. There are data suggesting that insertions and deletions are more immunogenic than missense mutations. Ideally, each given mutation deserves to be evaluated with regard to its potential immunogenicity within the individual HLA context. The mutation causes the immune response only if the affected gene is highly expressed, i.e., if it produces significant amounts of altered protein. The immune profile of the tumor tissue is also of high importance [80,91,92]. In summary, one may conclude that the clinical application of TMB is less defined as compared to the above-described MSI and HRD tests.

## 5. Conclusions and Perspectives

Integrative genomic tests are already widely utilized in clinical oncology. However, their intrinsic limitations deserve to be recognized. While there is actually no allowable room for imperfect reproducibility for “simple” mutation tests, e.g., EGFR, RAS, or BRAF assays, incomplete interlaboratory concordance for MSI, HRD, or TMB status determination cannot be avoided for the time being. Clinical validation of integrative genomic tests is significantly more challenging as compared to a single-gene analysis. While almost all EGFR/BRAF/ALK/ROS1/RET/MET-driven tumors respond to appropriate inhibitors, at least to some extent, currently applied MSI, HRD, and TMB assays are significantly less accurate in discriminating between drug responders and non-responders. Further development of complex genomic tests may involve the reconsideration and prioritization of the spectrum of analyzed DNA segments, with a focus on the most valuable genomic loci and elimination of insufficiently informative components of multigene assays, integration of these platforms with the expression analysis, and the search for user-friendly cost-efficient laboratory solutions.

## Figures and Tables

**Figure 1 ijms-23-13129-f001:**
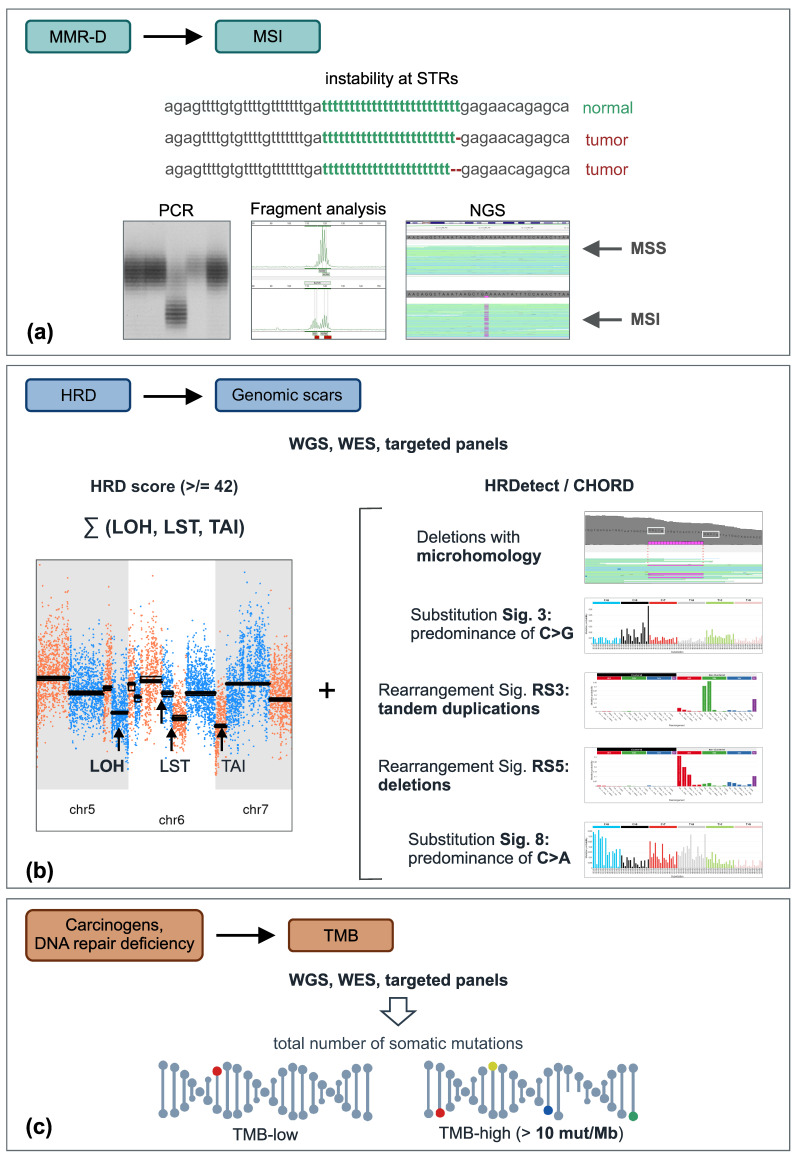
Integrative genomic tests. (**a**) Instability at short tandem repeats can be detected either by conventional PCR-based methods or by NGS. (**b**) Consequences of HR deficiency are usually determined by evaluating the level of chromosomal instability (e.g., HRD score (unweighted sum of LOH, LST, and TAI)) alone or in combination with specific mutational signatures. (**c**) TMB can be calculated from various NGS datasets, e.g., whole-genome, whole-exome, or targeted gene sequencing; colored circles designate various somatic mutations. Abbreviations: HRD, homologous repair deficiency; LOH, loss-of-heterozygosity; LST, large-scale state transition; MMR-D, mismatch repair deficiency; MSI, microsatellite instability; MSS, microsatellite stable; STR, short tandem repeats; TAI, telomeric allelic imbalance; TMB, tumor mutation burden; WES, whole-exome sequencing; WGS, whole-genome sequencing.

**Table 1 ijms-23-13129-t001:** Selected examples of single-gene assays and integrative genomic tests utilized in clinical oncology.

Genetic Alteration	Sensitivity to the Drugs/Other Applications	Tumor Types
**Single-gene tests**		
*BCR/ABL* fusions	ABL inhibitors (imatinib, dasatinib, nilotinib, bosutinib, ponatinib)	chronic myeloid leukemia, Ph+ acute lymphocytic leukemia
*EGFR* exon 19 and exon 21 mutations	EGFR inhibitors (gefitinib, erlotinib, afatinib, dacomitinib, osimertinib)/EGFR mutations in young-onset lung tumors are characteristic of Li-Fraumeni syndrome	lung cancer
*EGFR* exon 20 mutations	EGFR inhibitors (mobocertinib, amivantamab)	lung cancer
*HER2* exon 20 mutations	HER2 inhibitors (poziotinib)	lung cancer
*ALK* fusions	ALK inhibitors (crizotinib, ceritinib, brigatinib, alectinib, lorlatinib)	lung cancer, ALK+ anaplastic large cell lymphoma, inflammatory myofibroblastic tumors, other ALK-rearranged tumors
*ROS* fusions	ROS inhibitors (crizotinib, entrectinib)	lung cancer, other ROS1-rearranged tumors
*NTRK1/2/3* fusions	NTRK inhibitors (entrectinib, larotrectinib)	agnostic indication
*BRAF* mutations in codon 600	BRAF inhibitors	melanoma, colorectal cancer, lung cancer, hairy cell leukemia, thyroid cancer, other BRAF-mutated tumors
*RET* fusions or mutations	RET inhibitors (selpercatinib, pralsetinib)/RET mutation is an indicator of hereditary cancer if germline	lung cancer, thyroid cancer, other RET-driven tumors
*FGFR2/3* fusions or mutations	FGFR inhibitors (pemigatinib, infigratinib, erdafitinib)	urothelial cancer, cholangiocarcinoma
*KIT* mutations	KIT inhibitors (imatinib, dasatinib, nilotinib)	gastrointestinal stromal tumors (GIST), acral and mucosal melanomas
*MET* mutations	MET inhibitors (capmatinib, crizotinib)	lung cancer
*PIK3CA* mutations	PI3K inhibitors (alpelisib)	breast cancer
*IDH1/2* mutations	IDH inhibitors (ivosidenib, enasidenib)	acute myeloid leukemia
*BRCA1/2* mutations	DNA double-strand break-inducing drugs (cisplatin and other cytotoxic drugs, PARP inhibitors)/indicator of hereditary cancer if germline	breast cancer, ovarian cancer
**Integrative tests**		
Microsatellite instability	immune checkpoint inhibitors (pembrolizumab, nivolumab)/possible indicator of hereditary cancer	gastrointestinal cancers, endometrial cancers, other malignancies
Tumor mutation burden	immune checkpoint inhibitors (pembrolizumab, nivolumab)/ultra-high TMB is a possible indicator of hereditary cancer	approved as an agnostic indication; however, the list of malignancies needs to be further specified
Homologous repair deficiency	DNA double-strand break-inducing drugs (cisplatin and other cytotoxic drugs, PARP inhibitors)/possible indicator of hereditary cancer	breast cancer, ovarian cancer, prostate cancer

Notes: see [1,2,3] for references.

## Data Availability

Not applicable.

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
