# Peer review of "Integrative Genomic Tests in Clinical Oncology"

_ijms, 2022, doi:10.3390/ijms232113129_

Round 1
Reviewer 1 Report
The subject "Integrative Genomic Tests in Clinical Oncology" is a topical issue. The future of hope in cancer lies in biology/genetics to offer early diagnosis or molecular prognosis tests. This will allow a better understanding of cancer therapy.
Line 30: write in full DNA
Line40: "There are several medically important DNA tests which evaluate the entire land-40 scape of somatic mutations and provide grounds for the management of cancer disease".I would have liked to see the authors list these tests.
Line 44: "Examples of these tests include microsatellite instability 44 (MSI), tumor mutation burden (TMB), and homologous repair deficiency (HRD)." Your manuscript is well written, but I would have liked you to list in tables these different (current) tests or current studies on the issue.
Author Response
Comment: Line 30: write in full DNA
Response: It is highly uncommon to explain the abbreviation DNA in current scientific literature. In fact, the term “DNA” is perceived in the context of genetic code, while the term “deoxyribonucleic acid” is more associated with chemical properties of the substance. We suggest to abstain from the use of “deoxyribonucleic acid” in this paper, which is devoted to clinical significance of the analysis of nucleotide sequences.
Comment: Line 40: "There are several medically important DNA tests which evaluate the entire landscape of somatic mutations and provide grounds for the management of cancer disease". I would have liked to see the authors list these tests.
Response: We have incorporated Table 1 which lists all relevant tests.
Comment: Line 44: "Examples of these tests include microsatellite instability (MSI), tumor mutation burden (TMB), and homologous repair deficiency (HRD)." Your manuscript is well written, but I would have liked you to list in tables these different (current) tests or current studies on the issue.
Response: We have incorporated Table 1 which refers to these assays.
Reviewer 2 Report
The author sgive a gist of clinical gene tests for cancer which is well written. The way forward genetic assays/tests are well taken with three distinct sections. The Figure 1 is very good.
A few suggestions:
L103: Pl replace is with ARE
L319/363: immune therapy may be one word
L355: ARE relatively
L366: received A significant
L393: amountS
In Abstract L14/15: lack/require ( pl use singularis)
L31/32: The authors jump swiftly from pathogenic variants to drug. There could be missing link on genes harbouring pathogenic variants as targets.
L117: The authros could describe mutational PCR assays as well to check this
L147: Whence describing HRD, the authors could describe how advances in NGS for example ATAC-Seq is taking shape and bounds.
L160: Ex vivo assays are well taken and the authors could describe in few more sentences.
L325: multigene tests/panel developments/nanostring assays may be explained
There was no conclusions. Pl write few lines
Scores on a scale of 0-5 with 5 being the best
Language: 4.5
Novelty: 3.5
Brevity: 3.5
Scope/relevance: 4
Author Response
Comment: L103: Pl replace is with ARE
Response: “and the concordance of the results of MSI testing obtained by the conventional 5-marker panel and the MLH1/MSH2/MSH6/PMS2 IHC is close to 100%”. We believe that “is” is correct because it refers to “concordance” (singular, not plural).
Comment: L319/363: immune therapy may be one word
Response: We now use “immunotherapy” throughout the entire text.
Comment: L355: ARE relatively
Response: “However, its actual use in these tumor types is relatively uncommon.” We believe that “is” is correct because it refers to “its” (the use of TMB assay) (singular, not plural).
Comment: L366: received A significant
Response: Thank you, this is done.
Comment: L393: amountS
Response: Thank you, this is done.
Comment: In Abstract L14/15: lack/require (pl use singularis)
Response: We have re-phrased the sentence to meet this suggestion.
Comment: L31/32: The authors jump swiftly from pathogenic variants to drug. There could be missing link on genes harbouring pathogenic variants as targets.
Response: Thank you, we have inserted a linking phrase: “While normal DNA is examined in patients with suspected familial cancer syndromes, the identification of potential drug targets generally relies on the detection of somatic mutations in tumor DNA.”.
Comment: L117: The authors could describe mutational PCR assays as well to check this
Response: We have incorporated Table 1 in order to meet this suggestion.
Comment: L147: Whence describing HRD, the authors could describe how advances in NGS for example ATAC-Seq is taking shape and bounds.
Response: Thank you for this suggestion. We believe that the discussion on different varieties of NGS-based methods is outside the scope of this review. For example, ATAC-Seq has not been utilized for clinical determination of HRD. We kindly request to take our opinion with understanding.
Comment: L160: Ex vivo assays are well taken and the authors could describe in few more sentences.
Response: Thank you, we have some more explanations: “Homologous repair involves recruitment of RAD51 proteins to DNA double-strand breaks; consequently, cells with HRD do not demonstrate staining for RAD51-DNA complexes upon the induction of DNA lesions. RAD51 foci are intrinsically rare in tumors with low proliferation rate, therefore some RAD51 assays involve visual selection of mitotically active cells by geminin or Ki-67 staining [58].”
Comment: L325: multigene tests/panel developments/nanostring assays may be explained
Response: We believe that the discussion on multigene tests is an entirely different topic, which cannot be properly addressed within this review. We request not to incorporate this discussion in order to preserve the structure of the manuscript.
Comment: There were no conclusions. Pl write few lines
Response: Thank you very much! We have incorporated this section:
- Conclusions and perspectives
“Integrative genomic tests are already widely utilized in clinical oncology, however, their intrinsic limitations deserve to be recognized. While there is actually no allowable room for imperfect reproducibility for “simple” mutation tests, e.g., EGFR, RAS or BRAF assays, incomplete interlaboratory concordance for MSI, HRD or TMB status determination cannot be avoided for the time being. Clinical validation of integrative genomic tests is significantly more challenging as compared to a single-gene analysis. While almost all EGFR/BRAF/ALK/ROS1/RET/MET-driven tumors respond to appropriate inhibitors, at least to some extent, currently applied MSI, HRD and TMB assays are significantly less accurate in discriminating between drug responders and non-responders. Further development of complex genomic tests may involve reconsideration and prioritization of the spectrum of analyzed DNA segments, with a focus on the most valuable genomic loci and elimination of insufficiently informative components of multigene assays, integration of these platforms with the expression analysis, and the search for user-friendly cost-efficient laboratory solutions.”